# Addressing bladder and bowel challenges in dutch multiple sclerosis patients: Symptom prevalence and patient referral pathways

**Willemijn Faber[1]ⓔ, Alexander B. Stillebroer🆔[2]ⓔ***

**1** Department of rehabilitation, Rehabilitation Centre Heliomare, Wijk aan Zee, North Holland The Netherlands, **2** Department of Urology, Erasmus MC, Rotterdam, South Holland, The Netherlands

ⓔ These authors contributed equally to this work.
* a.stillebroer@erasmusmc.nl

## Abstract

### Objectives

Nerve damage and/or spinal cord injury in individuals with multiple sclerosis (MS) may result in the development of bladder and bowel problems. We wanted to gain insight into two issues: the prevalence of bladder and bowel problems among individuals with MS and the referral pathways for these problems to various physicians treating patients with MS.

### Methods

In the autumn of 2021, we undertook a nationwide cross-sectional study. We used a comprehensive questionnaire comprising 63 questions covering aspects such as MS diagnosis, disease progression, bladder and bowel issues, and consultations with healthcare professionals. Individuals with MS were invited to participate through social media or by direct mail.

### Results

The study included 1,082 validated respondents (81.5% female) with an average age of 53.3 years at participation, and an average age of 38.3 years at MS-diagnosis. Among the respondents, 91% experienced bladder problems while 73% indicated they had bowel problems. Within these respective groups, 42% and 47% were not aware of the connection between these issues and MS. Bladder and bowel related complaints disrupted daily life for 45% and 28% of the respondents, respectively. The patients consulted various healthcare providers for their complaints. Most respondents expressed dissatisfaction with the time it took to resolve their problems and 9% of those with bladder complaints and 60% of those with bowel complaints had not yet found a satisfactory solution.

### Conclusions

In the Netherlands, bladder and bowel symptoms in individuals with MS are highly prevalent and have a considerable impact on quality of life. Lack of awareness of these symptoms

**Data Availability Statement:** All data whereupon this manuscript is based can be accessed through this contact point: Secretary of the department of Urology of Erasmus MC, Dr. Molewaterplein 40,

3015 GD, Rotterdam, The Netherlands, email
Secretariaat.Urologie@erasmusmc.nl.

**Funding:** The author(s) received no specific
funding for this work.

**Competing interests:** Both authors (AS & WF) are
part of the Dutch MS advisory Board of Coloplast
BV. This does not alter our adherence to PLOS
ONE policies on sharing data and materials.

frequently leads to inadequate recognition, both by the patient and healthcare professional. Patients often consult various healthcare professionals for these complaints, which can potentially result in suboptimal treatment due to a lack of specialized expertise.

## Introduction

Multiple sclerosis (MS) is a chronic and frequently progressive autoimmune disease characterized by inflammation triggered by the immune system and demyelination of nerve fibers. This process results in brain and spinal cord damage, giving rise to a wide spectrum of symptoms [1]. It is estimated that approximately 2.8 million people worldwide live with MS [2], including approximately 34,000 in the Netherlands (Source: MS Association Netherlands). MS is more prevalent among women and typically emerges between the ages of 20 and 50 [3]. The disease manifests in various forms: benign MS, presenting mild symptoms; relapsing-remitting MS (RRMS), involving intermittent recovery; and primary and secondary progressive MS (PPMS and SPMS), characterized by a gradual worsening of symptoms [4]. The most prominent symptoms of MS include fatigue, cognitive impairments, and musculoskeletal issues.

MS-induced nerve and/or spinal cord damage can also lead to bladder and bowel dysfunction. A relationship exists between the degree of mobility impairment and the severity of bladder problems [5]. The most common bladder symptom is a sense of urinary urgency, with or without urge incontinence, stemming from reduced cortical control of the micturition reflex. Moreover, incomplete bladder emptying frequently occurs, due to 1) reduced detrusor muscle contractility stemming from decreased coordination by the pontine micturition center and 2) detrusor-sphincter dyssynergia, a functional obstruction due to urinary sphincter contraction during micturition [6]. Bowel dysfunctions arise from altered colon and/or anorectal function due to a lack of central control [7] including problems such as constipation, diarrhea, and fecal incontinence, among others. While bladder problems may occasionally lead to renal dysfunction, more often, urinary and fecal incontinence evoke feelings of shame, withdrawal from social interactions, and even social isolation. Our research aimed to shed light on the prevalence of bladder and bowel complaints in Dutch MS-patients and their corresponding referral pathways.

## Materials and methods

### Study design

In the autumn of 2021, we undertook a nationwide cross-sectional study. We used a comprehensive questionnaire comprising 63 questions covering aspects such as MS diagnosis, disease progression, bladder and bowel issues, and consultations with healthcare professionals.

The questionnaire incorporated questions derived from three validated questionnaires: 1) the Patient Determined Disease Steps (PDDS) questionnaire for MS symptomatology [8]; (2) the 8-item Short-form (SF)-Qualiveen questionnaire for bladder complaints [9]; and 3) the 10-item Neurogenic Bowel Dysfunction (NBD) questionnaire for bowel complaints [10]. The online survey was commissioned through an independent market research agency over a period of 6 weeks.

## Participation in the survey

Patients were invited to complete the online questionnaire via social media channels of the Dutch MS Association and the National MS Foundation or by direct mail for MS-patients registered in the database of Coloplast Netherlands BV.

Participants provided informed consent by ticking a box at the end of the questionnaire, stating that they had read the terms & conditions (their data was collected anonymously; according to GDPR and AVG guidelines and without the storage of an IP address–therefore, no individual answers or combinations could be retraced to specific participants). Only questionnaires completed by individuals with a diagnosis MS were included in the analysis. Questionnaires that were incomplete or completed in an unusual time frame ($< 7$ min or $>32$ min) were excluded.

Since participants were not subject to any violation of their physical and/or psychological integrity this study was not subject to review by an accredited medical ethical committee.

## Statistical analysis

We present the results in a descriptive manner. For this purpose, we used Microsoft 360 Excel and JASP v 0.61.4 (https://jasp-stats.org/download/, University of Amsterdam).

## Results

### Patients and disease characteristics

The online questionnaire was filled out by 1,161 partcipants; of them 1,082 consistently completed it. Mean completion time was 18 minutes (range: 7–32 minutes). The average age of the respondents was 53.3 years; they were diagnosed with MS at an average age of 38.3 years. Most respondents were women (81.5%), who were, on average, 3.6 years younger at the time of diagnosis compared to men. RRMS was the most frequent diagnosis (45.5%), followed by SPMS (21.1%), PPMS (19.9%), and mild/benign MS (6.2%). A higher proportion of men was diagnosed with PPMS compared to women (32.8% versus 17.0%), Conversely, RRMS was more frequent in women compared to men (49.5% versus 26.8%). Patients with PPMS and SPMS had the highest PDDS scores (Table 1).

The current average age and the age at diagnosis were comparable across disease profiles, except for a lower current age for respondents with RRMS (46.5 years) and a higher age at diagnosis for respondents with PPMS (46.6 years) (Table 2).

The average score for the SF-Qualiveen bladder symptom questionnaire was 1.20, and for the NBD bowel symptom questionnaire, it was 5.65.

Respondents who considered bladder or bowel symptoms as one of the top five limitations in their daily lives had higher average scores on both questionnaires. Among them, respondents with PPMS or SPMS recorded the highest scores on both questionnaires (Table 3).

Most respondents reported experiencing some form of bladder symptoms (91%) and bowel symptoms (73%). Among the respondents experiencing bladder and bowel symptoms, 42% and 47%, respectively, were unaware of any potential connection between these symptoms and MS. Bladder symptoms ranked among the top three limitations mentioned in the daily lives of respondents, with 45.3% reporting it alongside fatigue (67.2%), and a decline in cognitive abilities (42.1%) (Fig 1). A minority, but more than a quarter (27.9%) of respondents, identified bowel symptoms as a limitation in their daily lives.

Bladder and bowel symptoms were more of a limitation for men compared to women with 52.0% of men and 43.9% of women reporting limitations due to bladder symptoms, and 36.4% of men and 26.1% of women experiencing limitations from bowel symptoms. This trend

**Table 1. Demographic and disease characteristics.**

| | Male | Female | Unknown | Total |
|---|---|---|---|---|
| | **n = 198** | **n = 882** | **n = 2** | **N = 1,082** |
| **Gender** n, (%) | | | | |
| Male | 198 | | | 198 (18.3%) |
| Female | | 882 | | 882 (81.5%) |
| Unknown | | | 2 | 2 (0.2%) |
| **Current age** | | | | |
| Mean ± SD | 59.0 ±11.3 | 52.0 ±12.0 | 52.0 ±0.0 | 53.3 ± 12.1 |
| Median (min–max) | 60 (26–83) | 53 (16–80) | 52 (52–52) | 54 (16–83) |
| **Age at MS-diagnosis** | | | | |
| Mean ± SD | 41.2 ±11.8 | 37.6 ±10.9 | 46.0 ±4.2 | 38.3 ± 11.2 |
| Median (min–max) | 40 (18–68) | 37 (16–70) | 46 (43–49) | 38 (16–70) |
| **Stage of MS** (%) | | | | |
| Mild/benign | 12 (6.1) | 55 (6.2) | 0 | 67 (6.2) |
| Primary progressive | 65 (32.8) | 150 (17.0) | 0 | 215 (19.9) |
| Relapsing-remitting | 53 (26.8) | 437 (49.5) | 2 (100) | 492 (45.5) |
| Secondary progressive | 49 (24.7) | 179 (20.3) | 0 (0.0) | 228 (21.1) |
| Unknown | 19 (9.6) | 61 (6.9) | 0 (0.0) | 80 (7.4) |
| **PDDS total score** mean ± SD | | | | |
| Mild/benign | 2.5 ± 2.2 | 1.7 ± 1.5 | - | 1.8 ± 1.7 |
| Primary progressive | 4.8 ± 1.9 | 4.9 ± 1.8 | - | 4.9 ± 1.8 |
| Relapsing-remitting | 2.2 ± 1.8 | 2.3 ± 1.7 | 1 ± 0 | 2.3 ± 1.7 |
| Secondary progressive | 4.8 ± 2.0 | 4.7 ± 1.9 | - | 4.7 ± 1.9 |
| Unknown | 4.3 ± 2.4 | 3.0 ± 2.2 | - | 3.3 ± 2.3 |

Max: maximum, min: minimum, MS: multiple sclerosis, n: number of respondents, PDDS: Patient Determined Disease Steps, SD: standard deviation

[a]The PDDS comprises nine ordinal levels ranging between 0 (normal) and 8 (bedridden) [8].

**Table 2. Mean current age and mean age at MS-diagnosis per disease stage.**

| | Mild/ benign MS | PPMS | RRMS | SPMS | Unknown | Total |
|---|---|---|---|---|---|---|
| | **n = 67** | **n = 215** | **n = 492** | **n = 228** | **n = 80** | **N = 1,082** |
| **Current age** | | | | | | |
| Mean ± SD | 58.2 ± 11.1 | 60.8 ± 10.3 | 46.5 ± 10.7 | 58.1 ± 9.4 | 57.3 ± 11.3 | 53.3 ± 12.1 |
| min-max | 27–77 | 16–83 | 20–78 | 16–75 | 25–76 | 16–83 |
| **Age at MS-diagnosis** | | | | | | |
| Mean ± SD | 35.9 ± 10.2 | 46.6 ± 11.4 | 35.1 ± 9.7 | 38.4 ± 10.5 | 38.0 ± 10.8 | 38.3 ± 11.2 |
| min-max | 18–65 | 16–70 | 16–65 | 16–68 | 17–68 | 16–70 |

Max: maximum, min: minimum, MS: multiple sclerosis, PPMS: primary progressive MS, n: number of respondents, N: total number of respondents, RRMS: relapse remitting MS, SPMS: secondary progressive MS, SD: standard deviation

extended to limitations in sexual activities (31.3% versus 9.4%) and, to a lesser extent, mobility, muscle weakness, muscle stiffness, and hand function. Women identified sleep (20.8% versus 10.1%) and mood (10.4% versus 6.1%) as particularly significant limiting factors, and to a lesser extent fatigue, decline in cognitive abilities, change of mood, and pain.

**Table 3. SF-Qualiveen and neurogenic bowel dysfunction total score by disease stage for respondents who considered bladder or bowel complaints as one of the five most significant limitations in their daily life.**

| | Mild/ benign MS | PPMS | RRMS | SPMS | Unknown | Total |
|---|---|---|---|---|---|---|
| | n = 38 | n = 109 | n = 185 | n = 113 | n = 80 | N = 490 |
| **SF-Qualiveen total score[a]** | | | | | | |
| Median | 1.00 | 1.50 | 1.25 | 1.63 | 1.25 | 1.38 |
| Mean | 1.19 | 1.65 | 1.42 | 1.62 | 1.30 | 1.51 |
| SE of mean | 0.10 | 0.08 | 0.06 | 0.07 | 0.10 | 0.04 |
| SD of mean | 0.63 | 0.87 | 0.75 | 0.78 | 0.819 | 0.79 |
| Minimum | 0.25 | 0 | 0 | 0 | 0 | 0 |
| Maximum | 2.63 | 3.88 | 3.38 | 3.75 | 3.13 | 3.88 |
| **NBD score[b]** | | | | | | |
| Median | 2.50 | 5.00 | 4.00 | 5.00 | 5 | 5.00 |
| Mean | 3.76 | 6.97 | 6.38 | 7.11 | 6.71 | 6.62 |
| SE of mean | 0.57 | 0.60 | 0.44 | 0.63 | 0.77 | 0.28 |
| SD of mean | 3.52 | 6.28 | 5.93 | 6.71 | 6.87 | 6.26 |
| Minimum | 0 | 0 | 0 | 0 | 0 | 0 |
| Maximum | 13 | 28 | 30 | 39 | 38 | 39 |

NBD: Neurogenic Bowel Dysfunction, MS: multiple sclerosis, PPMS: primary progressive MS, RRMS: relapse remitting MS, SPMS: secondary progressive MS, SD: standard deviation, SE: standard error

[a]The Shortform (SF) Qualiveen comprises 8 questions; response options were presented on 5-point scales, where a score of 1 signifies no impact of urinary issues on Health-Related Quality of Life while a score of 5 indicates a significant adverse effect [9].

[b]The NBD comprises 10 multiple-choice questions with weighted answer options. Higher total scores are representing more severe bowel dysfunction (0–6 very minor, 7–9 minor, 10–13 moderate, and 14–47 severe) [10].

Urinary problems were reported less frequently as a limitation by respondents with RRMS (37.6%) compared to respondents with other forms of MS (49.6% to 56.7%). As for bowel problems, there were minimal distinctions reported among the different MS forms (Table 4).

## Consultations for bladder problems

Approximately one-third of respondents (34.2%) sought consultation with a healthcare professional within a year following the onset of bladder problems. A slightly higher percentage (37.5%) waited for over a year before seeking help, while 28.3% had not sought assistance at the time of the survey. Of the respondents 69.9% (n = 756) who discussed their bladder problems with a healthcare professional, 29.8% consulted a general practitioner, 29.0% a neurologist (29.0%), and 19.6% an MS nurse during their first visit (Fig 2A). Subsequent consultations varied (Fig 2B and 2C); for example, those who initially visited a general practitioner subsequently consulted a urologist (40.9%), a neurologist (19.1%), or an MS nurse (15.6%). Only a small number of respondents revisited the general practitioner for second or third consultations. After the first visit to the neurologist and MS nurse, most patients were referred to a urologist (55.7% and 37.2%, respectively). Following a second consultation with a neurologist, most respondents saw a urologist (56.2%) (Fig 2C).

## Consultations for bowel problems

Among respondents, 26.2% consulted a healthcare professional within a year after the onset of bowel problems, while 30.3% waited more than 1 year, and 43.3% had not sought help regarding this issue. Respondents who wanted to address their bowel problems (n = 620) primarily

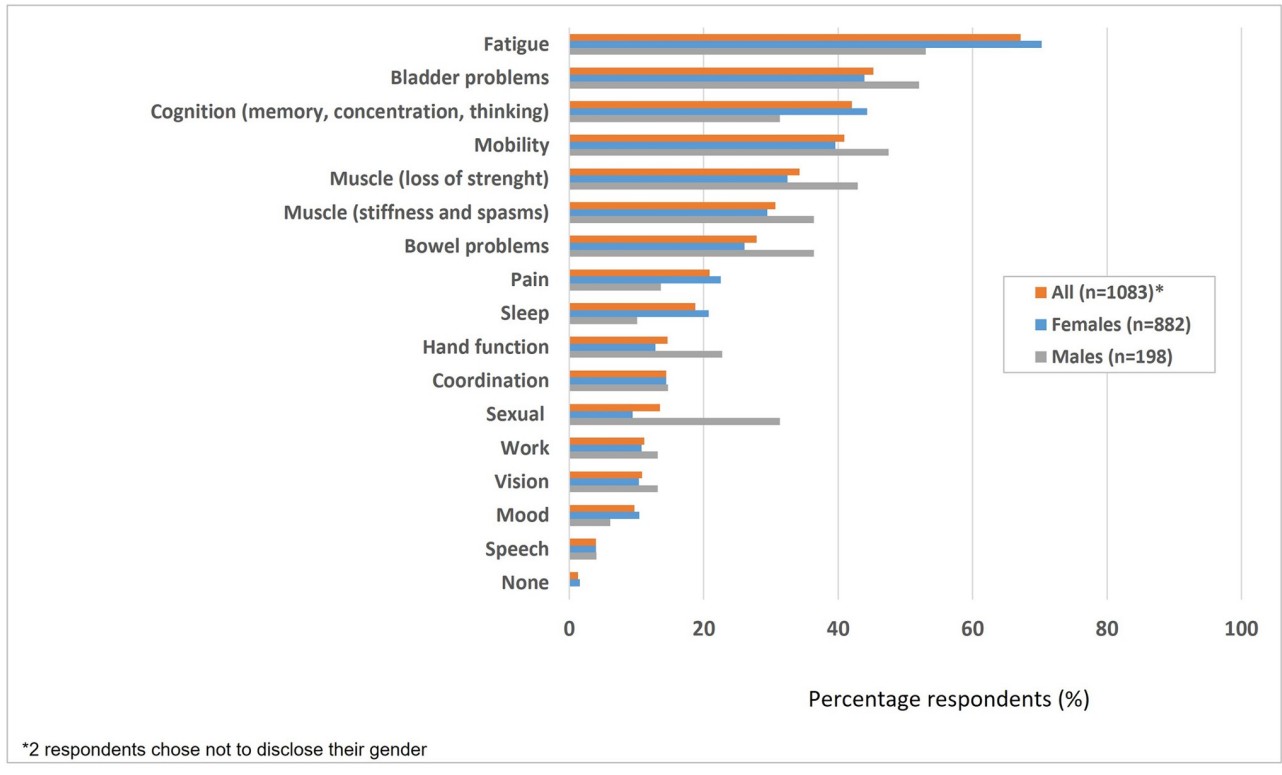

**Fig 1. Limiting factors in daily life in MS patients.**

consulted a general practitioner (38.7%) but also sought consultation from either a neurologist (17.1%) or an MS nurse (15.8%) (Fig 3A). Subsequent consultations involved multiple healthcare professionals. After the initial visit to the general practitioner, 25.8% of the patients saw a gastroenterologist, 16.3% consulted a neurologist, and 12.5% met with an MS nurse. Following the first consultation with an MS nurse, patients mainly consulted a neurologist (30.6%), and after the neurologist, they saw a gastroenterologist and an MS nurse (both 18.9%).

## Treatment of bladder symptoms

When asked about their satisfaction with the current solution or treatment of their bladder problems, a notable proportion (49.1%) of the 579 respondents indicated that they were either satisfied or very satisfied. Among them, 32.8% maintained a neutral perspective, 8.8% were dissatisfied or very dissatisfied, and 9.3% had not yet found a satisfactory solution. Regarding the time between the onset of the first bladder symptoms and the solution, 31.1% of

**Table 4. Experiencing bladder and bowel complaints as limiting, by disease stage.**

|  | Mild/ benign MS | PPMS | RRMS | SPMS |
|---|---|---|---|---|
|  | n = 38 | n = 109 | n = 185 | n = 113 |
| Bladder | 56.7% | 50.7% | 37.6% | 49.6% |
| Bowel | 26.9% | 27.4% | 25.6% | 32.0% |

MS: multiple sclerosis, PPMS: primary progressive MS, RRMS: relapse remitting MS, SPMS: secondary progressive MS

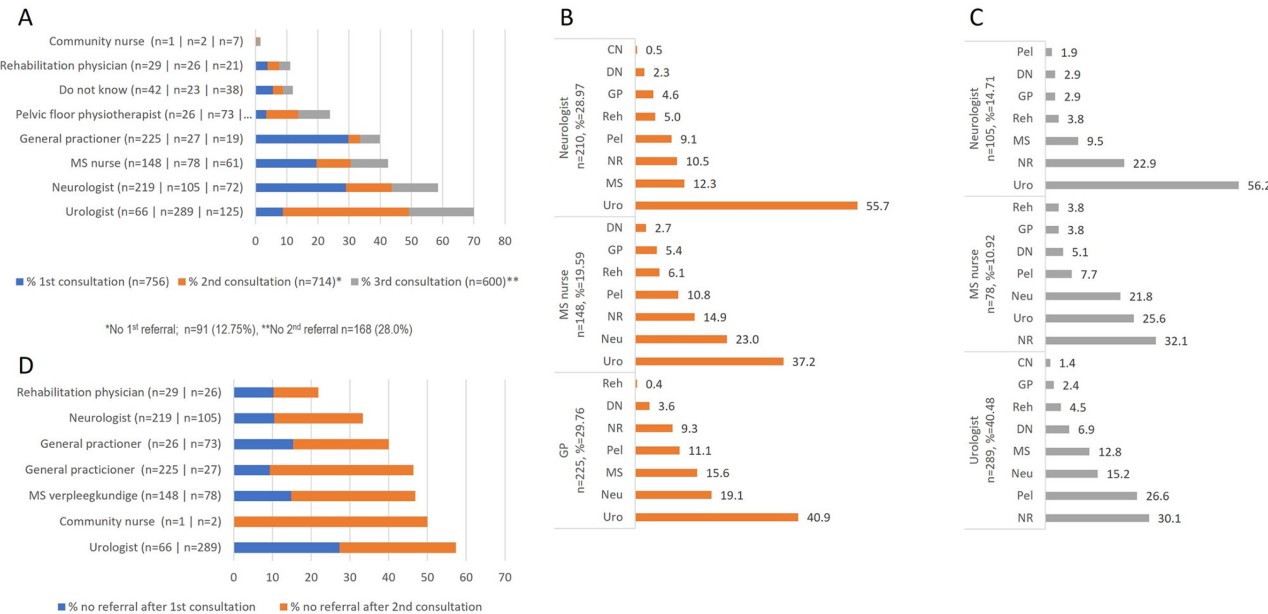

**Fig 2.** Consultations for bladder problems: (A) Percentage of respondents with a 1st, 2nd, and 3rd consultation per healthcare provider. (B) Percentage of respondents with a 2nd consultation following their 1st consultation with a neurologist, an MS nurse, and a general practitioner. (C) Percentage of respondents with a 3rd consultation following their 2nd consultation with a neurologist, an MS nurse, and a urologist. (D) Percentage of respondents who were not referred after the 1st or 2nd consultation per healthcare provider.

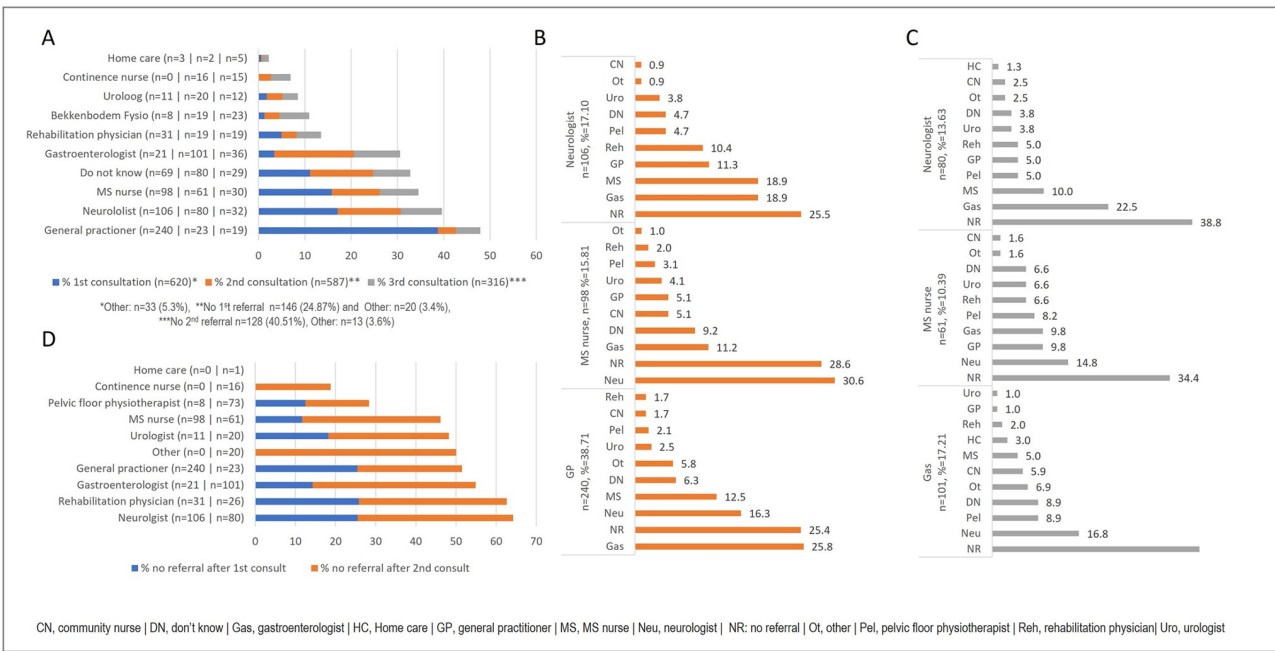

**Fig 3.** Consultations for bowel problems: (A) Percentage of respondents with a 1st, 2nd, and 3rd consultation per healthcare provider. (B) Percentage of respondents with a 2nd consultation following their 1st consultation with a neurologist, an MS nurse, and a general practitioner. (C) Percentage of respondents with a 3rd consultation following their 2nd consultation with the a neurologist, an MS nurse, and a gastroenterologist. (D) Percentage of respondents who were not referred after the 1st or 2nd consultation per healthcare provider.

respondents were satisfied or very satisfied, 55.0% were neutral, and 13.8% were dissatisfied or very dissatisfied. The respondents level of satisfaction showed little variation across different healthcare providers.

## Treatment of bowel symptoms

When asked about their satisfaction with the management of their bowel problems, a notable proportion (51.4%) of the 579 respondents indicated that they were dissatisfied; 59.9% found the solution insufficient. Of those respondents who found a solution, 41.3% were satisfied or very satisfied with the time between the onset of the initial bowel symptoms and the implementation of the solution, 39.6% were neutral, and 19.1% were dissatisfied or very dissatisfied. The respondents level of satisfaction showed little variation across different healthcare providers.

## Discussion

There is currently no published data from Dutch surveys documenting the prevalence of bladder and bowel problems among MS-patients. However, in clinical practice, these problems are frequently observed, as evidenced by our research findings (bladder issues: 91%; bowel issues: 73%). Drawing on international research, it is estimated that 87.6% of MS-patients experience bladder problems [11], while 39–75% experience bowel problems [7]. In a study in the US, 47% of patients with MS reported a significant impact of bladder problems in their daily life, while 28% experienced bowel problems [12].

A study with 1,100 MS patients showed that 14% had moderate to severe bowel dysfunctions (NBD score >10) [13], which is slightly lower compared to the 20% observed in our study.

Interestingly, a substantial proportion of the respondents were unaware that bladder (40%) and bowel (42%) problems could be related to their condition. When bladder problems are recognized as MS-related, they can be treated in accordance with the European [14] or US [15] Guideline for Neurogenic Bladder. Unfortunately, there are no international guidelines for the management of bowel problems associated with MS. It is worth noting that in the Netherlands, there are specialized centers with localized treatment protocols for managing these issues.

Our study demonstrates that patients frequently seek the input of various healthcare professionals for their concerns before finding satisfactory solutions. In the case of bladder problems, most patients ultimately consult a urologist, albeit often following a convoluted route. Our current study indicates that the time between diagnosis and treatment is often long and may not consistently lead to patient satisfaction. The fact that half of the respondents express satisfaction with the eventual treatment of their bladder problems and only a third are content with the time between the initial problems and the eventual solution suggests that the treatment pathway is not optimal.

In the case of bowel problems, patients often consult a gastroenterologist or MS nurse only after multiple referrals. This is consistent with the findings of Woodward et al [16] who concluded that variation in service provision depended on the knowledge and attitudes of HCPs, making referral pathways challenging.

In this regard, the treatment of bowel problems appears to be inadequate. Half of the respondents rate bowel management as 5 out of 10 or lower, and only 40% are satisfied with the eventual solution.

As shown by our study, most patients with MS initially consult their general practitioner regarding their bladder and/or bowel complaints. Given the size of the Dutch population, every general practitioner will, on average, have 2 to 3 MS-patients in their practice. With this in mind, it is strongly recommended that general practitioners recognize the potential

connection between these complaints and MS and refer patients to a healthcare professional with expertise in further diagnosing and treating these issues. In the case of bladder problems, we recommend referring patients to a urologist, and for bowel problems, directing them to specialized MS clinics or rehabilitation centers. It is important to recognize that bladder or bowel complaints may be initial symptoms of MS. Since many of these individuals initially present with these complaints to their general practitioner, it is crucial for general practitioners to be aware of the possible neurological origin of these complaints.

A substantial proportion of patients consult the MS nurse for their bladder and bowel complaints. Given their central role in the care of patients with MS, the MS nurse could play a pivotal role in this regard by increasing patient awareness about these MS-related complaints. Many patients tend to delay seeking help for these problems. Conducting a systematic assessment of bladder and bowel issues and having the MS nurse provide more precise referrals to the appropriate healthcare provider could improve this situation.

A strength of this survey is the participation of a substantial group of patients, exceeding 1,000 individuals, who completed the questionnaire in full. This suggests that the distribution networks used to conduct this survey reached a substantial number of MS-patients in the Netherlands (estimated 34,000).

However, it is important to acknowledge the possibility of selection bias, as an online survey will not reach all patients with MS in the Netherlands. Lack of digital literacy means that various patient groups may not have participated, a scenario that could have been addressed with a traditional written questionnaire. To mitigate this, we strategically employed diverse channels for survey distribution, including patient funds, patient associations, and industry, with the aim of reducing the impact of selection bias. Another potential concern is recall bias, especially when the number of respondents is relatively small. However, the respondents' characteristics in this survey align with the literature, particularly in the context of the management of bladder and bowel issues. Additionally, the large survey sample helps to minimize the risk of recall bias.

## Conclusion

The limited awareness among MS-patients of the connection between bladder and bowel issues and their condition hinders accurate recognition, both by the patients themselves and by healthcare professionals. The treatment path for these MS-related complaints is poorly defined, particularly in the case of bowel issues. As a consequence, patients turn to a range of healthcare professionals, resulting in less-than-optimal management of their bladder and bowel problems. Further research is imperative to develop an optimal treatment strategy for these problems in MS. Equally important is the education of healthcare professionals to enhance their ability to offer better support to patients with MS and bladder or bowel problems.

## Acknowledgments

We would like to express our gratitude to all the respondents for their participation in the study. We also extend our thanks to MarketResponse, MS Association Netherlands, the National MS Foundation in the Netherlands, and Coloplast Netherlands BV for their contributions to the research. Additionally, we acknowledge and are thankful for the editorial support provided by T4C, Hilversum, The Netherlands, in the preparation of this publication.

## Author Contributions

**Data curation:** Willemijn Faber, Alexander B. Stillebroer.

**Formal analysis:** Willemijn Faber, Alexander B. Stillebroer.

**Investigation:** Willemijn Faber, Alexander B. Stillebroer.

**Methodology:** Willemijn Faber, Alexander B. Stillebroer.

**Validation:** Willemijn Faber, Alexander B. Stillebroer.

**Writing – original draft:** Willemijn Faber, Alexander B. Stillebroer.

**Writing – review & editing:** Willemijn Faber, Alexander B. Stillebroer.

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
