## [Decision Letter · Decision Letter 0]

6 Sep 2024

Addressing Bladder and Bowel Challenges in Dutch Multiple Sclerosis Patients: Symptom Prevalence and Patient Referral Pathways

PONE-D-24-09995

Dear Dr. Stillebroer,

We’re pleased to inform you that your manuscript has been judged scientifically suitable for publication and will be formally accepted for publication once it meets all outstanding technical requirements.

Kind regards,

Gorica Maric

Academic Editor

PLOS ONE

“Both authors (AS & WF) are part of the Dutch MS advisory Board of Coloplast BV.”

Please respond by return email with your amended Competing Interests Statement and we will change the online submission form on your behalf.

3. In the online submission form you indicate that your data is not available for proprietary reasons and have provided a contact point for accessing this data. Please note that your current contact point is a co-author on this manuscript. According to our Data Policy, the contact point must not be an author on the manuscript and must be an institutional contact, ideally not an individual. Please revise your data statement to a non-author institutional point of contact, such as a data access or ethics committee, and send this to us via return email. Please also include contact information for the third party organization, and please include the full citation of where the data can be found.

Additional Editor Comments (optional): The present manuscript provides a comprehensive overview of current situation regarding bowel and bladder problems in MS population, covering all important aspects. Minor point - there is a typo in a line 237,  "With this is mind..." I suppose it should be "With this in mind".

Reviewers' comments:

Reviewer's Responses to Questions

**Comments to the Author**

1. Is the manuscript technically sound, and do the data support the conclusions?

Reviewer #1: Yes

2. Has the statistical analysis been performed appropriately and rigorously? 

Reviewer #1: Yes

3. Have the authors made all data underlying the findings in their manuscript fully available?

Reviewer #1: Yes

4. Is the manuscript presented in an intelligible fashion and written in standard English?

Reviewer #1: Yes

5. Review Comments to the Author

Reviewer #1: I believe this to be an important article given the lack of recognition in bladder and bowel issues in multiple sclerosis. The statistical analysis appears sound. The mean age of respondents is quite high I would have expected more respondents in a lower age range given the potential age of onset which may have altered order of patient 'complaints' to bring work and social activity issues higher in prioritisation. I feel there should also be some emphasis on the need for raising patient awareness of the issues related to bladder and bowel problems in the conclusion as patient activation, awareness and coping mechanisms here is very important.

6. PLOS authors have the option to publish the peer review history of their article (what does this mean?). If published, this will include your full peer review and any attached files.

Reviewer #1: No

---

## [Editor Report · Acceptance letter]

9 Oct 2024

PONE-D-24-09995 

PLOS ONE

Dear Dr. Stillebroer, 

I'm pleased to inform you that your manuscript has been deemed suitable for publication in PLOS ONE. Congratulations! Your manuscript is now being handed over to our production team.

Kind regards, 

on behalf of

Dr. Gorica Maric 

Academic Editor

PLOS ONE